# Protective Effect of Processed *Polygoni multiflori* Radix and Its Major Substance during Scopolamine-Induced Cognitive Dysfunction

**Ji-Hyun Kim [1], Ji Hyun Kim [2], Mei Tong He [1], Su Cheol Kim [2], Kyung Pan Hwa [2], Kye Man Cho [2] and Eun Ju Cho [1,*]**

[1] Department of Food Science and Nutrition & Kimchi Research Institute, Pusan National University, Busan 46241, Korea; llissunll@gmail.com (J.-H.K.); skyham16@gamil.com (M.T.H.)

[2] Department of Food Science, Gyeongnam National University of Science and Technology, Jinju 52725, Korea; jihyunkim@gntech.ac.kr (J.H.K.); dj055431@naver.com (S.C.K.); h9740013@naver.com (K.P.H.); kmcho@gntech.ac.kr (K.M.C.)

\* Correspondence: ejcho@pusan.ac.kr; Tel.: +82-51-510-2837

**Abstract:** Alzheimer's disease (AD) is the most common cognitive disorder in the elderly population. However, effective pharmacological agents targeting AD have not been developed. The processed *Polygoni multiflori* Radix (PPM) and its main active substance, 2,3,5,4′-tetrahydroxystilbene-2-*O*-β-glucoside (TSG), has received considerable attention, majorly due to its neuroprotective activities against multiple biological activities within the human body. In this study, we provide new evidence on the therapeutic effect of PPM and TSG during cognitive impairment by evaluating the ameliorative potential of PPM and TSG in scopolamine-induced amnesia in ICR mice. PPM (100 or 200 mg/kg) was orally administered during the experimental period (days 1–15), and scopolamine was intraperitoneally injected to induce cognitive deficits during the behavioural test periods (days 8–15). The administration of PPM and TSG significantly improved memory loss and cognitive dysfunction in behavioural tests and regulated the cholinergic function, brain-derived neurotrophic factor, and neural apoptosis. The present study suggests that PPM and TSG improved scopolamine-induced cognitive dysfunction, but further study has to be supported for the clinical application of PPM and TSG for AD prevention and treatment.

**Keywords:** processed *Polygoni multiflori* Radix; 2,3,5,4′-tetrahydroxystilbene-2-O-β-glucoside; Alzheimer's disease; cognitive dysfunction





## 1. Introduction

With the increase in the ageing population, the number of people living with dementia has been increasing. In 2015, the number of people living with dementia was estimated to be 46.8 million worldwide, which is estimated to rise to 74.7 million in 2050 [1]. Alzheimer's disease (AD) is the most common cognitive disorder among neurodegenerative dementia and is a major contributor to disability and dependence in the elderly population [2]. AD is distinguished from other forms of dementia by characteristic changes in the brain. In AD, there is a cerebral accumulation of plaque deposits of the β-amyloid peptide (Aβ) and tau protein aggregation into neurofibrillary tangles [3]. Another characteristic feature of AD is reduced production of certain brain chemicals necessary for communication between nerve cells, such as acetylcholine (ACh), norepinephrine, serotonin, and somatostatin [4]. In addition, AD causes progressive impairment of behavioural and cognitive functions, including memory loss and executive and visuospatial dysfunction [5]. Nonetheless, there is a paucity of highly effective drugs that prevent cognitive dysfunction and behavioural symptoms of AD [6]. Recent studies have demonstrated that active substances in plants can be used to potentially establish novel therapies for AD [7]. Compounds isolated

from plants, such as flavonoids and other polyphenols in medicinal plants, might prevent neurodegeneration and enhance memory and cognitive function [8]. Thus, products from certain plants are a major source for new drug discovery.

*Polygoni multiflori* Radix (PM) originates from the root of *Polygonum multiflorum* Thunb., which has been used as a tonifying traditional medicine and a dietary supplement in East Asia and North America [9]. The role of PM has been investigated in various biological activities, including antioxidant activity, nerve cell protection, lipid regulation, and hair-follicle growth [10,11]. Furthermore, PM has been used for a long time as an anti-ageing agent [12]. However, the adverse effects of PM, such as hepatotoxicity, skin allergy, and bleeding in the upper digestive tract, have been reported occasionally [13,14]. Abundant evidence has shown that processed PM (PPM), boiled in black bean liquid according to a traditional procedure, is efficient in reducing the toxicity of PM. Hence, PPM is considered relatively safe and has attracted increasing attention as a promising therapeutic agent instead of PM [13,15]. Moreover, 2,3,5,4′-tetrahydroxystilbene-2-*O*-β-glucoside (TSG), a monomer of stilbene, is an important component isolated from PPM [16]. Therefore, we used TSG in the present study as a standard constituent of PPM.

In a previous study, PM exhibited a protective effect on $A\beta_{25-35}$-induced cognitive deficits [17]. Although the positive effect of PM on $A\beta_{25-35}$-induced cognitive deficits has been demonstrated, there are no studies on the cognitive enhancement activities of PPM, particularly in scopolamine-induced cognitive dysfunctions. Scopolamine is an anticholinergic drug, and it impairs short-term learning and memory functions, which are AD-like symptoms in mice [18]. Scopolamine-induced have mice shown cognitive and memory dysfunction by cholinergic dysfunction and neuronal apoptosis [18,19]. Therefore, scopolamine-induced mice have been used for AD mice models to investigate the protective effect of plants and their active compounds on AD. This study aimed to investigate the improvement effect of PPM and its major substance, TSG, on cognitive impairment using novel object recognition (NOR), passive avoidance (PA), and Morris water maze (MWM) tests. In addition, we investigated the changes in acetylcholinesterase (AChE) activity, acetylcholine (ACh) concentration, and protein levels of brain-derived neurotrophic factor (BDNF), B-cell lymphoma 2 (Bcl-2), and Bcl-2-associated X (Bax) to confirm the neuroprotective mechanisms of PPM and TSG in scopolamine-induced AD-like dysfunctions.

## 2. Materials and Methods

### 2.1. Reagents

The seven standards of phytochemicals, including catechin, TSG, emodin-8-*O*-β-D-glucopyranoside (EG), questin, rhein, emodin, and chrysophanol, were obtained from Sigma-Aldrich Inc. (St. Louis, MO, USA). Scopolamine, DNP, acetylcholinesterase activity assay kit, and 2,7-dichloro-fluorescein diacetate (DCF-DA) were purchased from Sigma-Aldrich Inc. Sodium chloride (NaCl) was obtained from Lps Solution Co. Ltd. (Daejeon, Korea). The acetylcholine assay kit was supplied by Abcam Plc. (Cambridge, UK). Phosphoric acid was purchased from Samchun Pure Chemical Co. Ltd. (Seoul, Korea).

### 2.2. Preparation of PPM

The PPM powder was purchased from Jirisan-hasuo Farming Corporation (Sancheong, Gyeongsangnam-do, Korea). The PM was washed three times, and then the washed PM was steamed for 60 min at 95 °C ± 2 °C. The steamed contents were put into an ageing container where they were allowed to age at 75 °C ± 2 °C for 72 h. This process was repeated three times. The PPM was placed in a dry oven at 50 °C ± 2 °C for 72 h, which allowed the residual water to evaporate [20]. Thereafter, it was crushed into a powdered form using a grinder. Then, 10 g of PPM powder was added to 200 mL of 50% EtOH, and the mixture was extracted at 70 °C ± 2 °C for 8 h. The extract was filtered through a 0.45-μm membrane filter to recover the supernatant. This process was repeated twice. To prepare the extract concentrates, each extract supernatant was concentrated to approximately 12 °Bx (Brix = soluble solid contents) using a rotary evaporator. The extract

concentrates and diluent samples were tested to evaluate their phytochemical compounds and protective effects.

### 2.3. Analysis of Phytochemical Compounds

The analysis of phytochemical compounds was modified and performed as previously described using high-performance liquid chromatography (HPLC, Agilent 1260 series, Agilent Co., Santa Clara, CA, USA) methods of Jung et al. [20]. For the analysis of the seven phytochemicals, one gram of PPM powders was added to 20 mL of the 50% ethanol, and the mixtures were extracted at 70 ± 2 °C for 8 h. The extract samples were filtered through a 0.45 μm membrane filter for analysis of HPLC. Samples were separated on an XTerra RP18 analytical column (4.6 × 250 mm, 5 μm; Waters Corp., Milford, MA, USA). The mobile phase was composed of 0.5% glacial acetic acid in water (solution A) and solvent B (0.5% glacial acetic acid in acetonitrile) using the following gradient program: 0 min 0% B, 10 min 15% B, 15 min 5% B, 20 min 15% B, 25 min 5% B, 30 min 10% B, 40 min 50% B, 45 min 60% B, 60 min 90% B, and 65 min 100% B. The injection volume was 20 μL, and the flow rate was 1.0 mL/min at 30 °C (oven temperature). The detection wavelength was UV 280 nm.

### 2.4. Animals and Treatment

Five-week old male ICR mice weighing 25–28 g were purchased (Orient Inc., Seongnam, Korea). The mice were housed in individual cages under controlled temperature (20 °C ± 2 °C), 12-h light/dark cycle, and specific humidified conditions (50% ± 10%). All mice were provided ad libitum access to water and a standard diet (5L79, Orient, Korea). The animal protocol used in the present study was approved by the Pusan National University-Institutional Animal Care and Use Committee (PNU-IACUC; Approval Number PNU-2020-2670). After one week of acclimatisation, the ICR mice (*n* = 56) were randomly divided into seven groups (*n* = 8): normal: no-treatment group; control: scopolamine-treated group; PPM100 or PPM200: PPM (100 or 200 mg/kg)-treated scopolamine group; TSG10 or TSG50: TSG (10 or 50 mg/kg)-treated scopolamine group; and DNP: DNP (5 mg/kg)-treated scopolamine group. PPM, TSG, and DNP were dissolved in distilled water to a suitable concentration, respectively, and administered daily via gastric gavage. DNP was used as a positive control. Cognitive dysfunction was induced by a single scopolamine injection (1.5 mg/kg, intraperitoneal (i.p.)). The behavioural experiments were conducted 30 min after scopolamine injection. The experimental design employed in this study is illustrated in Figure 1.

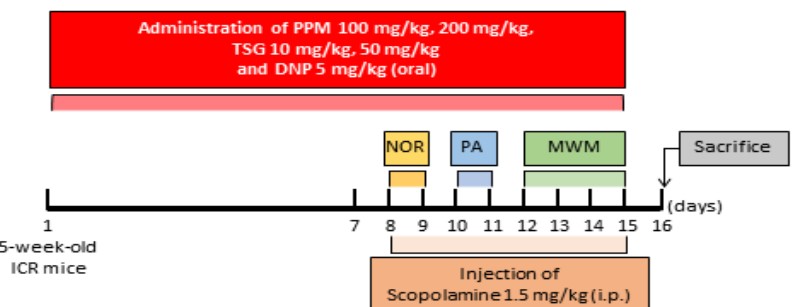

**Figure 1. Schedule of the behavioural experiment.** Normal = distilled water; control = scopolamine 1.5 mg/kg; PPM100 = scopolamine 1.5 mg/kg + PPM 100 mg/kg; PPM200 = scopolamine 1.5 mg/kg + PPM 200 mg/kg; TSG10 = scopolamine 1.5 mg/kg + TSG 10 mg/kg; TSG50 = scopolamine 1.5 mg/kg + TSG 50 mg/kg; DNP = scopolamine 1.5 mg/kg + DNP 5 mg/kg. PPM, processed *Polygoni multiflori* Radix; TSG, 2,3,5,4′-tetrahydroxystilbene-2-*O*-β-glucoside; DNP, donepezil hydrochloride; NOR, novel object recognition; PA, passive avoidance; MWM, Morris water maze.

*2.5. Behavioural Studies*

2.5.1. Novel Object Recognition Test

Object discrimination ability was observed using the NOR test, as described previously [21]. The experimental set up (apparatus) consisted of a rectangular box (40 × 40 × 40 cm) made of a black polyester plastic chamber. The test consisted of two sessions: training and test sessions conducted over a period of two days. On the training day, the mice were individually allowed to explore two identical objects for 10 min inside the apparatus. On the day of the test, each mouse was replaced in the apparatus in which one of the identical objects was changed with a novel object, which is a different shape and colour from familiar objects [22]. Exploratory behaviour was defined only as sniffing or touching the object within 1 cm. The apparatus was cleaned using 70% ethanol to avoid olfactory cues after every trial, and the next mouse was placed inside the apparatus. Specific parameters including the number of touches on the familiar ($F$) object, the number of touches on the novel ($NN$) object, and the total number of touches on both the objects ($N + F$) were individually evaluated. The object recognition ability (%) was calculated using the following formula:

$$\text{Object recognition ability } (\%) = \frac{F \text{ or } N}{N + F} \times 100$$

2.5.2. Passive Avoidance Test

The PA test was performed to evaluate learning and memory parameters for fear in scopolamine-treated mice, as described in a previous study [23]. The PA test apparatus was divided into an illuminated compartment and a dark compartment separated by a guillotine door. Briefly, during the training trial, the mice were placed in the illuminated compartment, and when the mice entered the dark compartment, a mild electric shock (0.5 mA, 3 s duration) was applied. After 24 h, the mice were placed again in the illuminated compartment, and 30 s later, the guillotine door was opened. The latency time in which the mice tried to remain in the illuminated compartment was recorded as the retention time. If the mice did not enter the dark compartment, the upper limit was set to 300 s.

2.5.3. Morris Water Maze Test

The long-term and spatial memory was determined using the MWM test as described previously [24]. The water maze apparatus consisted of a circular-shaped pool (diameter: 95 cm, height: 45 cm). The pool was filled with black-dyed opaque water (temperature: 22 °C ± 2 °C) and divided into four equal quadrants with visual cues attached to them. An escape platform (diameter: 8 cm) was set 1 cm beneath the water surface in the fixed quadrant centre. The MWM test was conducted as a training session (three times a day, at a four-hour interval) over three consecutive days, followed by a probe day as the testing session. In the training session, the mice were randomly placed in one quadrant and released into the pool. The swimming time to find the platform was recorded. If the mice found the platform within 60 s, the mice remained for an additional 10 s. If the mice did not find the platform within 60 s, the swimming time was recorded as 60 s, and the experimenter guided the mice to the platform leaving it for 10 s to recognise the location. In the testing session, three different trials for other purposes were evaluated at four-hour intervals. First, the time that arrived at the escape platform was observed. Subsequently, the escape platform was removed, the time spent in the target quadrant, and the time taken to find the exposed platform were measured and analysed. All swim path lengths were recorded using a computerised video-tracking system (Panlab, Barcelona, Spain).

*2.6. AChE Activity and ACh Concentration in Brain*

All mice were anaesthetized with a zoletil$^{50}$ and rumpun mixture (3:1 ratio) and sacrificed for sample collection after completing the behavioural tests. The brains were carefully dissected and stored at −80 °C immediately. The supernatant of the brain homogenate was

used to determine AChE activity and ACh concentration using a commercially available kit according to the manufacturer's instructions. The absorbance of AChE activity and ACh concentration was evaluated using a FluoStar Optima plate reader (BMG Labtech, Ortenberg, Germany).

*2.7. Western Blot Analysis*

The brain tissues were homogenised on ice with lysis buffer containing a protease inhibitor cocktail. The homogenate was centrifuged at 13,000 rpm for 30 min at 4 °C, and the protein concentration was evaluated using the dye concentrate (Bio-Rad, Richmond, CA, USA). Equal amounts of protein samples were separated by sodium dodecyl sulphate polyacrylamide gel electrophoresis (SDS–PAGE) and transferred to polyvinylidene fluoride membranes (Millipore, Burlington, MA, USA). The membranes were blocked for 60 min using 10% skimmed milk in phosphate-buffered saline (PBS) containing 0.5% Tween 20 (PBS-T) and then incubated with specific primary antibodies (BDNF (1:1000, Abcam plc), Bcl-2 (1:1000, Abcam plc), Bax (1:1000, Cell Signalling Technology), and β-actin (1:200, Santa Cruz)) overnight at 4 °C. After complete washing with PBS with 0.1% Tween 20 (PBST), the membrane was incubated with HRP-conjugated secondary antibodies (1:1000) at room temperature for another 60 min. The protein bands were visualised using picoenhanced peroxidase detection (ELPIS-Biotech, Daejeon, Korea). Western blot bands were detected using a Davinci-Chemiluminescent imaging system (CoreBio, Seoul, Korea).

*2.8. Statistical Analysis*

All data are expressed as the mean ± standard deviation (SD). The search error data were analysed using one-way analysis of variance (ANOVA) followed by Duncan's multiple range test using SPSS software (version 20.0, IBM Corporation, Armonk, NY, USA). The results from the NOR and PA tests in the behavioural experiments were demonstrated using a Student's *t*-test. The differences were considered statistically significant when $p < 0.05$.

## 3. Results

*3.1. Phytochemical Contents of PPM*

The typical HPLC chromatograms of phytochemical peaks obtained from PPM (Figure 2) and their contents are shown in Table 1. According to HPLC analysis, the contents of catechin, TSG, EG, and emodin were 0.43, 18.81, 0.23, and 0.21 mg/g, respectively. The TSG compound accounted for approximately 96% of PPM. Several researchers have reported that the TSG compound accounts for >95% of PPM [16,18].

**Table 1.** Phytochemical compounds of processed *Polygoni multiflori* Radix.

| Phytochemical Compounds | Contents [1] (mg/g) |
|---|---|
| Catechin | 0.43 ± 0.03 [c] |
| 2,3,5,4′-tetrahydroxystilbene-2-*O*-β-glucoside (TSG) | 18.81 ± 0.75 [a] |
| Emodin-8-*O*-β-D-glucopyranoside (EG) | 0.23 ± 0.02 [b] |
| Questin | nd [2] |
| Rhein | nd |
| Emodin | 0.21 ± 0.01 [b] |
| Chrysophanol | nd |

[1] All values are presented as the mean ± SD of pentaplicate determination. Different letters (a–c) indicate significant differences ($p < 0.05$) by Duncan's multiple range test. [2] nd: not detected.

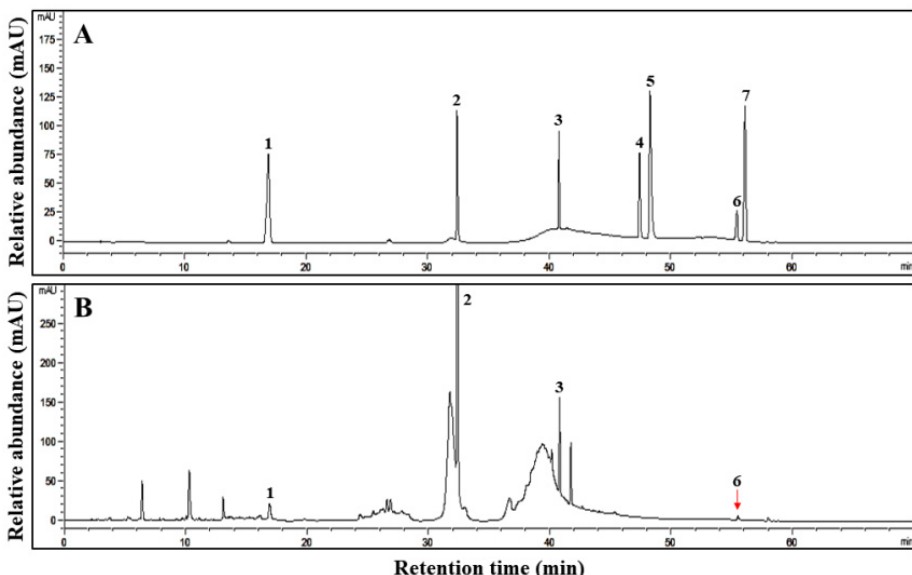

**Figure 2.** Typical HPLC chromatograms of the 50% EtOH extract from processed *Polygoni multiflori* (PPM) (**A**) standards and (**B**) PPM. 1, catechin; 2, TSG; 3, emodin-8-*O*-β-D-glucopyranoside (EG); 4, questin; 5, rhein; 6, emodin; and 7, chrysophanol.

### 3.2. PPM and TSG Ameliorate Scopolamine-Induced Cognitive Dysfunctions in Behavioural Tests

3.2.1. NOR Test

The results of the NOR test are shown in Figure 3A. In the testing session, statistical analyses demonstrated that scopolamine-administered mice showed significant memory deficits. There were no significant differences between familiar and novel objects in the exploration percentage of scopolamine-administered control mice. However, administration of PPM (100 and 200 mg/kg), TSG (10 and 50 mg/kg), and DNP (5 mg/kg) elevated the object discrimination ability similarly to that of normal mice. The administration of PPM and TSG significantly improved short-term memory and object discrimination ability in scopolamine-treated mice.

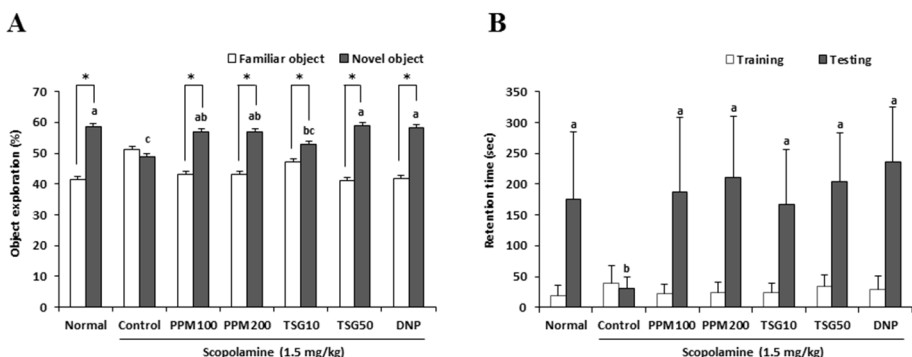

**Figure 3.** Effect of PPM and TSG on scopolamine-induced cognitive dysfunction in the NOR test and PA test. The results of the NOR test (**A**) and PA test (**B**) are shown. Normal = distilled water; control = scopolamine 1.5 mg/kg; PPM100 = scopolamine 1.5 mg/kg + PPM 100 mg/kg; PPM200 = scopolamine 1.5 mg/kg + PPM 200 mg/kg; TSG10 = scopolamine 1.5 mg/kg + TSG 10 mg/kg; TSG50 = scopolamine 1.5 mg/kg + TSG 50 mg/kg; DNP = scopolamine 1.5 mg/kg + DNP 5 mg/kg. Values are the mean ± SD (n = 8). * $p < 0.05$ compared with the old route by Student's *t*-test. Different letters (a–c) indicate significant differences ($p < 0.05$) by Duncan's multiple range test.

### 3.2.2. PA Test

The results obtained from the PA test are shown in Figure 3B. During the training trial, all groups showed a non-significant behaviour pattern in their latency time. In the testing trial, the scopolamine injection significantly reduced the latency time to 31 s, compared to the normal group, which displayed a latency time of 175 s. Interestingly, administration of PPM and TSG alleviated memory retention, with significant amelioration in latency time, compared to the scopolamine-injected control mice. At the higher dose of PPM (200 mg/kg) and TSG (50 mg/kg), the latency time was reversed to 210 and 203 s, respectively, near the DNP group at 235 s. These results suggest that PPM and TSG improved learning and memory abilities in scopolamine-treated mice.

### 3.2.3. MWM Test

The effects of PPM and TSG on long-term and spatial memory were measured using the MWM test. As shown in Figure 4, the swimming time for mice to find the escape platform decreased in all groups except the scopolamine-induced mice during the entire experimental period. The scopolamine-treated mice in the testing session exhibited a significantly longer latency time than the other groups. These results demonstrated that scopolamine triggered the impairment of long-term and spatial memory. In comparison, the administration of PPM and TSG showed a shorter latency time compared with that of the scopolamine-injected mice. In particular, PPM (100 and 200 mg/kg) induced similar effects to those of the normal and DNP groups. The swimming routes further verified that scopolamine-induced learning and memory impairment was attenuated by PPM and TSG treatment (Supplemental Figure S1). Additionally, PPM- and TSG-administered mice stayed for a longer time in the target quadrant after removing the platform (Figure 5). These results collectively demonstrated that PPM and TSG protected the disruption of long-term and spatial memory retention induced by scopolamine. Meanwhile, the latency time to find the exposed platform did not differ among all groups (Supplemental Figure S2), indicating that all experimental mice's visual and swimming abilities did not affect cognitive function.

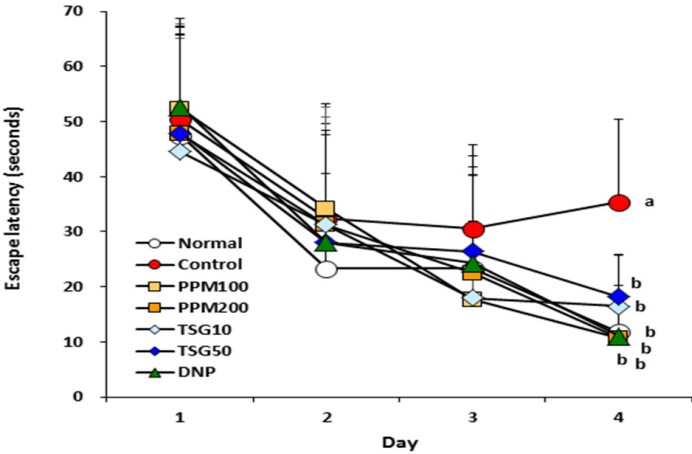

**Figure 4.** Escape latency to reach the hidden platform in scopolamine-injected mice during the MWM test. Normal = distilled water; control = scopolamine 1.5 mg/kg; PPM100 = scopolamine 1.5 mg/kg + PPM 100 mg/kg; PPM200 = scopolamine 1.5 mg/kg + PPM 200 mg/kg; TSG10 = scopolamine 1.5 mg/kg + TSG 10 mg/kg; TSG50 = scopolamine 1.5 mg/kg + TSG 50 mg/kg; DNP = scopolamine 1.5 mg/kg + DNP 5 mg/kg. Values are the mean $\pm$ SD (n = 8). Different letters (a,b) indicate significant differences ($p < 0.05$) by Duncan's multiple range test.

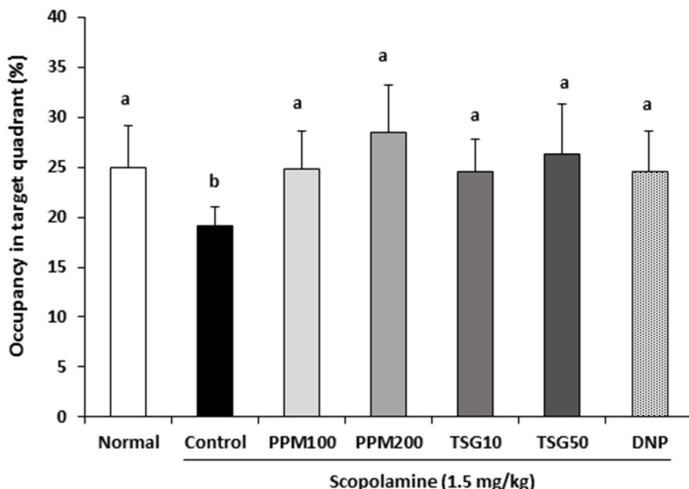

**Figure 5.** Occupancy in target quadrant of scopolamine-injected mice during MWM test. Normal = distilled water; control = scopolamine 1.5 mg/kg; PPM 100 = scopolamine 1.5 mg/kg + PPM 100 mg/kg; PPM200 = scopolamine 1.5 mg/kg + PPM 200 mg/kg; TSG10 = scopolamine 1.5 mg/kg + TSG 10 mg/kg; TSG50 = scopolamine 1.5 mg/kg + TSG 50 mg/kg; DNP = scopolamine 1.5 mg/kg + DNP 5 mg/kg. Values are the mean $\pm$ SD (n = 8). Different letters (a,b) indicate significant differences ($p < 0.05$) by Duncan's multiple range test.

*3.3. PPM and TSG Attenuate Scopolamine-Induced Cholinergic System Dysfunctions in the Brain*

To identify the influence of PPM and TSG on scopolamine-induced cholinergic system dysfunction, we evaluated the indices of the cholinergic system (i.e., AChE and ACh activities) in the brain (Figure 6A,B). Scopolamine injection significantly increased AChE activity in the brain relative to the normal group. However, PPM and TSG significantly inhibited the scopolamine-induced increase in AChE activity in the brain. Moreover, the concentrations of ACh in scopolamine-induced mice were reduced remarkably in the brain. In contrast, PPM and TSG reversed the decrease in ACh concentration in scopolamine-injected mice brain. These data suggest that PPM and TSG ameliorated scopolamine-induced dysfunction of the cholinergic system in the brain. Moreover, improvement observed in behavioural deficits with the use of PPM and TSG when induced by scopolamine could be associated with the control of AChE and ACh.

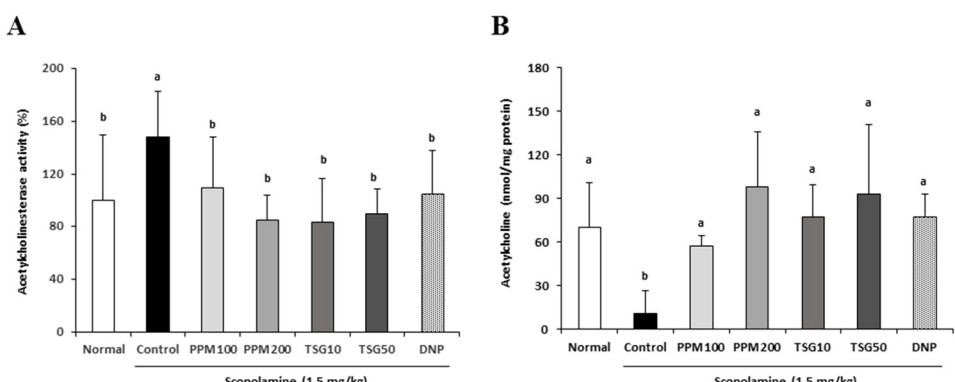

**Figure 6.** Effect of PPM and TSG on scopolamine-induce cholinergic dysfunction in the brain. AChE activity (**A**) and ACh concentration (**B**) are shown. Normal = distilled water; control = scopolamine 1.5 mg/kg; PPM100 = scopolamine 1.5 mg/kg + PPM 100 mg/kg; PPM200 = scopolamine 1.5 mg/kg + PPM 200 mg/kg; TSG10 = scopolamine 1.5 mg/kg + TSG 10 mg/kg; TSG50 = scopolamine 1.5 mg/kg + TSG 50 mg/kg; DNP = scopolamine 1.5 mg/kg + DNP 5 mg/kg. Values are the mean $\pm$ SD (n = 8). Different letters (a,b) indicate significant differences ($p < 0.05$) by Duncan's multiple range test.

### 3.4. PPM and TSG Upregulate BDNF Expression in Scopolamine-Induced Brain

In Figure 7, we show the effect of PPM and TSG on BDNF expression in the scopolamine-induced brain. Scopolamine-injected mice showed suppressed BDNF expression compared to the normal group. In contrast, PPM and TSG administration significantly elevated BDNF protein levels in a dose-dependent manner. In particular, high doses (PPM, 200 mg/kg; TSG, 50 mg/kg) of the PPM and TSG groups upregulated the protein level of BDNF that was comparable with that in the DNP group, suggesting that PPM and TSG effectively enhanced neuronal function that was damaged by scopolamine.

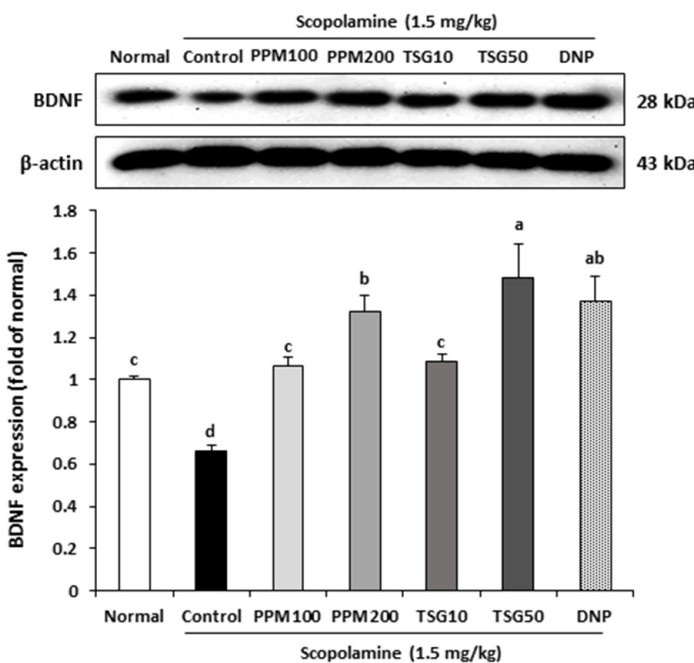

**Figure 7.** Effect of PPM and TSG on BDNF expression in scopolamine-induced brain. Normal = distilled water; control = scopolamine 1.5 mg/kg; PPM 100 = scopolamine 1.5 mg/kg + PPM 100 mg/kg; PPM200 = scopolamine 1.5 mg/kg + PPM 200 mg/kg; TSG10 = scopolamine 1.5 mg/kg + TSG 10 mg/kg; TSG50 = scopolamine 1.5 mg/kg + TSG 50 mg/kg; DNP = scopolamine 1.5 mg/kg + DNP 5 mg/kg. Values are the mean ± SD (n = 8). The different letters (a–d) indicate significant differences ($p < 0.05$) by Duncan's multiple range test.

### 3.5. PPM and TSG Inhibit Scopolamine-Induced Neural Apoptosis in the Brain

The ratio of Bax to Bcl-2 expression in the brain was examined to determine whether PPM and TSG could inhibit neural apoptosis induced by scopolamine (Figure 8A). In the scopolamine-injected group, Bax expression was significantly increased and Bcl-2 expression was significantly decreased in the brain compared with the normal group (Figure 8B,C). Hence, the ratio of Bax to Bcl-2 was remarkably higher than that of the scopolamine-injected control group. However, the increase in the ratio of Bax to Bcl-2 expression was inhibited upon administration of PPM and TSG. A high dose (TSG, 50 mg/kg) of TSG group reinstated the ratio of Bax to Bcl-2 expression to a level similar to that in the normal group. These findings proved that PPM and TSG effectively inhibited scopolamine-induced neural apoptosis in the brain.

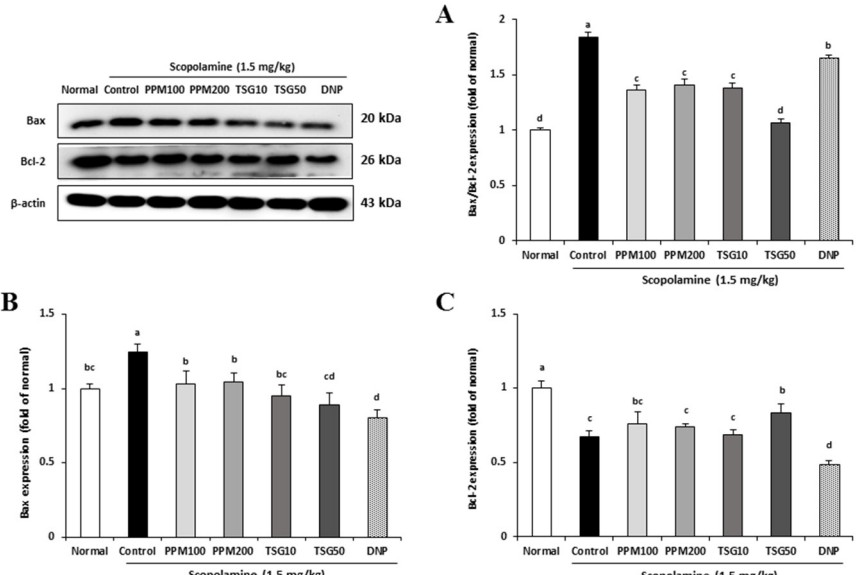

**Figure 8.** Effect of PPM and TSG on scopolamine-induced neuronal apoptosis in the brain. The ratio of Bax/Bcl-2 expression (**A**), Bax expression (**B**), and Bcl-2 expression (**C**) are shown. Normal = distilled water; control = scopolamine 1.5 mg/kg; PPM 100 = scopolamine 1.5 mg/kg + PPM 100 mg/kg; PPM200 = scopolamine 1.5 mg/kg + PPM 200 mg/kg; TSG10 = scopolamine 1.5 mg/kg + TSG 10 mg/kg; TSG50 = scopolamine 1.5 mg/kg + TSG 50 mg/kg; DNP = scopolamine 1.5 mg/kg + DNP 5 mg/kg. Values are the mean ± SD (n = 8). The different letters (a–d) indicate significant differences ($p < 0.05$) by Duncan's multiple range test.

## 4. Discussion

AD is caused by multiple aetiologies, such as cholinergic dysfunction, neuronal apoptosis, oxidative stress, and neuroinflammation, in the body [25]. The scopolamine-induced mice model is widely used to evaluate the effectiveness on AD in the preclinical test. In addition, it provides the results of the promising agents on cognitive functional and its mechanisms under in vivo preclinical test [18]. Therefore, further study has to be supported on cognitive improvement and its mechanisms of PPM or TSG in the clinical study on the basis of the results of scopolamine-induced mice models. In the present study, we investigated the behavioural improvement and protective mechanisms of cognitive impairment of PPM and its major constituent, TSG. The chemical profiles demonstrated that the major characteristic constituents of PPM are TSG, emodin, and physcion [26]. These compounds are extensively assumed to be responsible for the bioactivities of PPM. Numerous studies in the field of pharmacology have verified that PPM and its active constituents could enhance immune functions and elicit anti-oxidative and anti-ageing activities [27]. Among the active constituents of PPM, TSG has been shown to cross the blood–brain barrier and has protective effects on hippocampal synaptic plasticity [28]. Numerous studies have recently indicated that TSG alleviates cognitive dysfunction and inhibits ROS generation in several animal models of AD, including $A\beta_{1-42}$-injected rats and APP transgenic mice [29,30]. Based on these studies, we planned the overall experimental groups, including TSG, as an index component of PPM. Moreover, DNP was approved under Food and Drug Administration (FDA) drugs for the treatment of AD, and it is a well-known treatment agent for AD [31]. DNP acts as a cholinesterase inhibitor by enhancing cholinergic neurotransmission [31]. A number of studies reported that cognitive improvement effects of DNP were related to the regulation of cholinergic dysfunction, neuronal apoptosis, and others, in scopolamine-induced cognitive impairment mice [32,33]. However, DNP has several side effects, such as vomiting, diarrhoea, nausea, and muscle cramps in the body [34]. Therefore, many studies have been focused on the development of natural products without side effects that can replace synthetic drugs for the treatment of AD [35]. In this study, we investigated the cognitive improvement effects of PPM and

TSG, which is an index component of PPM. The DNP was used as a positive control for the evaluation of cognitive function.

AD is a multifactorial neurodegenerative brain disease that causes behavioural and cognitive dysfunction in ageing adults [36]. The key neuropathological mechanisms of AD are surmised to be associated with cholinergic damage, oxidative stress, and neuronal loss-mediated neurodegeneration [37]. Therefore, medicine candidates targeting these multiple causes could be a promising strategy to suppress the progression of AD. The present study investigated the cognitive enhancement effect of PPM and TSG in a scopolamine-induced AD mouse model. Administration of PPM and TSG significantly improved scopolamine-induced cognitive impairment, cholinergic function, and oxidative stress.

Cholinergic malfunction leads to the deterioration of learning and memory processing in AD patients [38]. Scopolamine, a cholinergic blocking agent, is a standard drug to provoke cognitive failure in healthy humans, and it is commonly used to produce a pathologic model for AD/amnesia [18]. In the current study, we utilised a mimic AD model, a scopolamine-injected mice model, to investigate the anti-AD activity of PPM and TSG. The experimental results showed that scopolamine injection (1.5 mg/kg, i.p.) successively induced cognitive dysfunction in ICR mice. Behavioural experiments in animal models have been conducted to study the effectiveness of PPM and TSG for AD. We carried out behavioural tests, including the NOR, PA, and MWM tests. These behavioural tests were confirmed to investigate memory loss for deficits in object discrimination (NOR), learning memory (NOR, PA, MWM), and spatial memory (MWM).

The NOR test is mainly used for studying both short- and long-term memory [39]. Accelerated memory loss is a pathologic condition that occurs in AD [40]. The NOR test was performed by measuring the preference for the novel object over the familiar one. Object recognition memory is related to the integrity of the hippocampus and perirhinal cortex [41]. Scopolamine impaired the object discrimination ability, as evidenced by the decreased exploration percentage in the novel object. However, the PPM and TSG treatment significantly increased the exploration percentage in the novel object compared to the familiar object and comparable to DNP. The increase in novel object exploration percentage in mice indicated alleviation of learning and memory impairment by scopolamine. This suggests that PPM and TSG possess memory-enhancing activity.

The animal in a PA test learns that a specific place should be avoided when related to a bothering incident, such as an electric shock. Mice naturally prefer dark sections, but the mice lose this tendency upon receiving electric shock [42]. The PA test is an evaluation of cognitive memory based on the avoidance of a fear-inducing condition. In scopolamine-treated mice, the avoidance memory was disrupted, and the retention time to stay in the bright section after the electric shock was shortened. However, the shortened step-through latency was prolonged in PPM- and TSG-treated mice. This result was similar to that of DNP-treated mice, demonstrating a successful acquisition of avoidance memory. PPM and TSG prevented impairment in avoidance memory during scopolamine-induced cholinergic receptor blockade conditions.

The MWM test is frequently used to monitor spatial learning and memory in mice, which offers highly reliable information on cognitive evaluation [43]. Since the mice have to swim in a large round pool to reach a hidden platform, they learn the positional relation between the platform and landscape and arrive at the platform more quickly. Therefore, we can assume their spatial learning ability from escape latency. For these advantages, this method is among the most popular behavioural tests to assess spatial learning and memory ability [44]. In this investigation, relative to scopolamine-treated mice, the PPM and TSG administration ameliorated the search error during the training period, suggesting an improvement in learning and memory function impairment. Additionally, based on scopolamine-treated mice's latencies, the PPM and TSG administration elevated the time spent in the target cue quadrant when the hidden platform was removed. Taken together with behavioural analyses and the results of NOR, PA, and MWM tests, PPM and TSG

administration improved memory and learning regarding object discrimination, avoidance memory, and spatial memory in the scopolamine-induced AD-like mice.

Scopolamine primarily blocks ACh receptors and subsequently induces cognitive dysfunction [45]. ACh, an essential neurotransmitter in the central nervous system, plays an important role in learning and memory retention by regulating the cholinergic system. Studies have reported that patients with AD show cholinergic deficits (i.e., elevated AChE activity and reduced Ach content) [46]. The present study demonstrated that scopolamine treatment increased AChE activity and decreased ACh content in the mouse brain. On the other hand, PPM and TSG treatment inhibited AChE activity and increased ACh concentration. The DNP-treated mice exhibited a similar effect to that of PPM-and TSG-treated mice. In scopolamine-injected control mice, AChE rapidly hydrolysed ACh, and the suppression of AChE could elevate the synaptic concentration of ACh. However, PPM and TSG-treated mice affected the cholinergic function. Therefore, PPM and TSG could possess neuroprotective effects on scopolamine-induced cognitive dysfunction through functional enhancement of the cholinergic system.

BDNF regulates neuronal survival, growth, and release of neurotransmitters. In the mammalian neocortex, BDNF plays an important role in the function of neurons, including glutamatergic, GABAergic, serotonergic, dopaminergic, adrenergic, and cholinergic neurons [47]. Particularly, several studies have reported that higher BDNF levels, together with ACh levels, are essential for synaptic plasticity and cognitive behaviour, and there are strong positive correlations between ACh and BDNF in the brain [48]. Additionally, the protein level of BDNF is impaired in patients with AD as well as in scopolamine-induced animal models. Hence, BDNF is a vital modulator in memory formation [49]. The BDNF has two forms, the precursor (pro BDNF) and its mature form of BDNF in the brain. The precursor BDNF preferentially binds to the p75 neurotrophin receptor (NTR), and then it converts to mature BDNF cleaved by extracellular proteases [50]. The mature BDNF preferentially binds to tropomyosine-related kinase B (TrkB) receptor [50]. Pro BDNF binds to p75 NTR, and then activates apoptotic pathways in neurons and glial, and it negatively regulates neuronal remodelling and synaptic plasticity [51,52]. On the other hand, mature BDNF enhances long-term potentiation, and it is critical for neuroplasticity and neuronal function in the brain [51]. In the AD patients, both pro BDNF and mature BDNF were decreased in the brain [53]. In our study, administration of the PPM or TSG group elevated protein levels of mature BDNF in the brain compared with the scopolamine-induced control group, indicating an amelioration effect for the retention and storage of memory on cognitive impairment induced by scopolamine. According to a previous study, TSG-administered mice increased BDNF as well as TrkB expression in the brain [54]. A previous study investigated the effects of PM on pro-BDNF and mature BDNF expression in neurotoxicity-induced hippocampal neuronal cells [55]. The treatment of PM significantly increased mature BDNF expression [55]. Therefore, TSG could activate the mature form of BDNF, but the study of BDNF processing and signalling of PPM or TSG is necessary to use PPM or TSG as therapeutic materials for AD.

Apoptosis, programmed cell death, is among the key mechanisms underlying neuronal survival. The anti-apoptotic Bcl-2 and pro-apoptotic Bax are two pivotal molecules related to apoptosis, and the ratio of Bax to Bcl-2 determines if cells would undergo apoptosis [56]. The results of the present study showed that neuronal apoptosis, as indicated by the ratio of Bax to Bcl-2 expression in the brain, was significantly increased in the scopolamine-injected control group compared with the normal group. In contrast, PPM and TSG treatment revoked the changes. These results suggest that the improvement effect of PPM and TSG on scopolamine-induced cognitive dysfunction is correlated with preventing apoptotic neuronal death.

In the present study, PPM, TSG, and DNP significantly improved cognitive function by regulations of cholinergic dysfunction, BDNF expression, and neuronal apoptosis in scopolamine-induced cognitive impairment mice. Several previous studies also demonstrated that the administration of TSG at doses of 20, 40, and 80 mg/kg/day improved

cognitive function, such as fear memory and novel object recognition by the upregulation of BDNF in normal mice [57]. In addition, the treatment of TSG significantly increased the expression of neurotrophic factors, including BDNF, nerve growth factor (NGF), and glial cell-derived neurotrophic factor (GDNF) in rat primary astroglial cells [58]. Furthermore, the administration of TSG increased hippocampal synaptic plasticity in normal mice [59]. These previous studies support the cognitive improvement effects of TSG and PPM under normal mice models as well as AD mice models.

Taken together with our results, the administration of PPM and its index component, TSG, improved cognitive function in the behaviour tests, and they had similar effects to the administration of DNP. In addition, PPM- and TSG-administered groups attenuated cholinergic dysfunction, similarly to the DNP-administered group. The administration of PPM and TSG increased BDNF expression similarly to the DNP-administered group. Furthermore, PPM- and TSG-administered groups at all concentrations significantly attenuated neuronal apoptosis by the reduction of the Bax/Bcl-2 ratio, compared with the DNP-administered group.

## 5. Conclusions

In conclusion, the present study confirmed that PPM and TSG could reverse scopolamine-induced cognitive dysfunction, as highlighted by NOR, PA, and MWM tests in mice. Moreover, our results provide evidence that the cognitive enhancement effects of PPM and TSG might be linked to anti-cholinergic effects, improvement of neuron function, and anti-apoptotic events within the brain. Therefore, we suggest that PPM and TSG improve scopolamine-induced cognitive dysfunction in mice, but further study has to be supported for the clinical application of PPM and TSG for AD prevention and treatment.

**Supplementary Materials:** The following are available online at https://www.mdpi.com/2227-9717/9/2/342/s1, Figure S1: Swimming paths in scopolamine-injected mice during the MWM test; Figure S2: Visual and physical activities during the MWM test.

**Author Contributions:** Conceptualization, J.H.K. and E.J.C.; formal analysis, K.M.C.; investigation, J.-H.K., M.T.H. and S.C.K.; resources, K.P.H. and K.M.C.; data curation, J.-H.K. and E.J.C.; writing—original draft preparation, J.-H.K.; writing—review and editing, J.H.K., K.M.C. and E.J.C.; supervision, E.J.C. All authors have read and agreed to the published version of the manuscript.

**Funding:** This research received no external funding.

**Institutional Review Board Statement:** The study was approved by the Pusan National University-Institutional Animal Care and Use Committee (PNU-IACUC; Approval Number PNU-2020-2670).

**Informed Consent Statement:** Not applicable.

**Data Availability Statement:** Not applicable.

**Acknowledgments:** This work was supported by "Food Functionality Evaluation program" under the Ministry of Agriculture, Food and Rural Affairs and partly Korea Food Research Institute [G202200-01].

**Conflicts of Interest:** The authors declare no conflict of interest.

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
