# Peer review of "Protective Effect of Processed Polygoni multiflori Radix and Its Major Substance during Scopolamine-Induced Cognitive Dysfunction"

_processes, doi:10.3390/pr9020342_

Round 1

Reviewer 1 Report

Dear Authors,  

the article entitled »Protective Effect of Processed Polygoni Multiflori Radix and Its Major Substance during Scopolamine-Induced Cognitive Dysfunction« provides new evidence on therapeutic effect of Polygony multiflori Radix and 2,3,5,4′-tetrahydroxystilbene-2-O-β-glucoside in scopolamine-induced amnesia in ICR mice.

The results presented are important and should be published. Minor corrections were added to the text. My major concerns are the HPLC method that should be described in more detail and the lack of normal distribution evaluation for statistical tests. Moreover, the relation between AD and scopolamine-induced cognitive dysfunction should be described in more detail in the introduction.

Overall, mine suggestion is that the manuscript would be acceptable after minor revision.

Author Response

Thank you for the valuable comments on this paper. We considered the comments carefully and the manuscript has been revised according to the comments.

The results presented are important and should be published. Minor corrections were added to the text.

; We revised minor corrections in the manuscript.

[Minor corrections in PDF file]

Can you explain in more detail the relationship between scopolamine-induced cognitive dysfunction and AD-
; We explained it in introduction (Line 64-69, Page 2).

[Introduction]

Scopolamine is an anticholinergic drug and it impairs short-term learning and memory functions, which are AD-like symptoms in the mice (Alvarez-Jimenez et al., 2016). Scopolamine-induced mice showed cognitive and memory dysfunction by cholinergic dysfunction and neuronal apoptosis (Alvarez-Jimenez et al., 2016; Sohn et al., 2019). Therefore, scopolamine-induced mice has been used for AD mice model to investigate the protective effect of plants and its active compounds from AD.

[References]

Alvarez-Jimenez, R.; Groeneveld, G.J.; van Gerven, J.M. A.; Goulooze, S.C.; Baakman, A.C.; Hay, J.L.; Stevens, J. Mod-el-based exposure-response analysis to quantify age related differences in the response to scopolamine in healthy sub-jects. Br. J. Clin. Pharmacol. 2016, 82, 1011-1021. https://doi.org/10.1111/bcp.13031.

Sohn, E.; Lim, H.S.; Kim, Y.J.; Kim, B.Y.; Jeong, S.J. Annona atemoya leaf extract improves scopolamine-induced memory impairment by preventing hippocampal cholinergic dysfunction and neuronal cell death. Int. J. Mol. Sci. 2019, 20, 3538. https://doi.org/10.3390/ijms20143538.

Why did you chose this method? If it is possible add a reference.
; Thanks for the good suggestion. It was rewritten as follows (Line 88-93, Page 2):

[2.2. Preparation of PPM]

The PPM powder was purchased from Jirisan-hasuo Farming Corporation (Sancheong, Gyeongsangnam-do, Korea). The PM was washed three times and then the washed PM was steamed for 60 min at 95°C ± 2°C. The steamed contents were put in an ageing container where they were allowed to age at 75°C ± 2°C for 72 h. This process was repeated three times. The PPM was placed in a dry oven at 50°C ± 2°C for 72 h, which al-lowed the residual water to evaporate (Jung et al., 2020).

[References]

Jung, S.; Son, H.; Hwang, C.E.; Cho, K.M.; Park, S.W.; Kim, H.; Kim, H.J. The root of Polygonum multiflorum Thunb. alleviates non-alcoholic steatosis and insulin resistance in high fat diet-fed mice. Nutrients 2020, 12, 2353. https://doi.org/10.3390/nu12082353.

The test used require normal distribution. Did you test for it?
; Thanks for the good advice. It was rewritten as follows (Line 103-115, Page 3):

The analysis of phytochemical compounds was modified and performed as previously described using a high-performance liquid chromatography (HPLC, Agilent 1260 series, Ag-ilent Co., Santa Clara, CA, USA) methods of Jung et al. (2020). For analyzed the seven phytochemicals, one gram of PPM powders was added the 20 mL of the 50% ethanol and the mixtures was extracted at 70±2˚C for 8 h. The extract samples were filtered through a 0.45-µm membrane filter for analysis of HPLC. Samples were separated on an XTerra RP18 analytical column (4.6 × 250 mm, 5 µm; Waters Corp., Milford, MA, USA). The mobile phase was composed of 0.5% glacial acetic acid in water (solution A) and solvent B (0.5% glacial acetic acid in acetonitrile) using the following gradient program: 0 min 0% B, 10 min 15% B, 15 min 5% B, 20 min 15% B, 25 min 5% B, 30 min 10% B, 40 min 50% B, 45 min 60% B, 60 min 90% B, and 65 min 100% B. The injection volume was 20 µL, and the flow rate was 1.0 mL/min at 30°C (oven temperature). The detection wavelength was UV 280 nm.

[2.3. Analysis of phytochemical compounds]

; We added and written as follows in Table 1:

1All values are presented as the mean ± SD of pentaplicate determination. Different letters (a–c) indicate significant differences (p < 0.05) by Duncan’s multiple range test.

[References]

Jung, S.; Son, H.; Hwang, C.E.; Cho, K.M.; Park, S.W.; Kim, H.; Kim, H.J. The root of Polygonum multiflorum Thunb. alleviates non-alcoholic steatosis and insulin resistance in high fat diet-fed mice. Nutrients 2020, 12, 2353. https://doi.org/10.3390/nu12082353.

Please comment the relation between mice and humans. Could be the protective effect be different in human compared to mice?

; We explained it in discussion (Line 371-377, Page 10)
[Discussion]

AD is caused by multiple etiologies such as cholinergic dysfunction, neuronal apoptosis, oxidative stress, neuroinflammation, and the others, in the body [Qiu et al., 2009]. The scopolamine-induced mice model is widely used to evaluate the effectiveness on AD in the preclinical test. In addition, it provides the results of the promising agents on cognitive functional and its mechanisms under in vivo preclinical test [Al-varez-Jimenez et al., 2016]. Therefore, further study has to be supported on cognitive improvement and its mechanisms of PPM or TSG in the clinical study on the basis of results on scopolamine-induced mice model.

[References]

Qiu, C.; Kivipelto, M.; von Strauss, E. Epidemiology of Alzheimer's disease: occurrence, determinants, and strategies toward intervention. Dialogues Clin. Neurosci. 2009, 11, 111-128. https://doi.org/10.31887/DCNS.2009.11.2/cqiu.

Alvarez-Jimenez, R.; Groeneveld, G.J.; van Gerven, J.M. A.; Goulooze, S.C.; Baakman, A.C.; Hay, J.L.; Stevens, J. Mod-el-based exposure-response analysis to quantify age related differences in the response to scopolamine in healthy sub-jects. Br. J. Clin. Pharmacol. 2016, 82, 1011-1021. https://doi.org/10.1111/bcp.13031.

My major concerns are the HPLC method that should be described in more detail and the lack of normal distribution evaluation for statistical tests.

; Thank you comment. This comment is described above.

Moreover, the relation between AD and scopolamine-induced cognitive dysfunction should be described in more detail in the introduction.

; We explained it in introduction (Line 64-69, Page 2).

[Introduction]

Scopolamine is an anticholinergic drug and it impairs short-term learning and memory functions, which are AD-like symptoms in the mice (Alvarez-Jimenez et al., 2016). Scopolamine-induced mice showed cognitive and memory dysfunction by cholinergic dysfunction and neuronal apoptosis (Alvarez-Jimenez et al., 2016; Sohn et al., 2019). Therefore, scopolamine-induced mice has been used for AD mice model to investigate for plants and its active compounds for treatment of AD.

[References]

Alvarez-Jimenez, R.; Groeneveld, G.J.; van Gerven, J.M. A.; Goulooze, S.C.; Baakman, A.C.; Hay, J.L.; Stevens, J. Mod-el-based exposure-response analysis to quantify age related differences in the response to scopolamine in healthy sub-jects. Br. J. Clin. Pharmacol. 2016, 82, 1011-1021. https://doi.org/10.1111/bcp.13031.

Sohn, E.; Lim, H.S.; Kim, Y.J.; Kim, B.Y.; Jeong, S.J. Annona atemoya leaf extract improves scopolamine-induced memory impairment by preventing hippocampal cholinergic dysfunction and neuronal cell death. Int. J. Mol. Sci. 2019, 20, 3538. https://doi.org/10.3390/ijms20143538.

Reviewer 2 Report

The data present in this manuscript are interesting for understanding the potential of plant-derived chemicals in improving the cognitive disorder. The reviewer only has a couple of major questions about the relative role of real acting chemicals (e.g. TSG and DNP) in PPM.

In the present work, the authors looked into the effect of PPM, pure TSG and DNP on cognitive behavior (measured by several cognitive related tests) of mice that have been treated with scopolamine. The results indeed clearly showed that the positive effect of improving the cognitive dysfunctions caused by scopolamine. To be accurate, these data only demonstrated that PPM TSG or DNP (at certain doses) can improve the cognitive dysfunctions induced by scopolamine. However, one cannot definitively conclude that PPM could be a functional food and an effective dietary supplement for the prevention and treatment of AD. Such type of conclusion is a bit overstated, thus need to be be revised to a moderate tone.  Additionally, the authors have not studied the difference in cognitive response between non-treated normal group and only PPM (or TSG, or DNP)-treated group. Such type of experimental results certainly help convince the role of PPM or TSG or DNP in improving the cognitive  behavior, this can be one drawback for well understanding the aim of this work, and thus, the discussion is essentially required.

Secondly, what is the relative roles of TSG and DNP? Dose the promising therapeutic potential of PPM correlate with TSG or DNP? This is addressed in the discussion.  

Minor points:

“To prepare the extract concentrates, each extract supernatant was concentrated to approximately 12 °Bx using a rotary evaporator.” What is the meaning of 12 Bx?

For the phytochemical analysis, the flow rate, injection volume, column (XTerra™ RP18, 4.6 × 250 mm, 5 μm, Waters Crop., Milford, MA, USA), and oven temperature were adjusted at 1 mL/min, 20 μL, UV 280 nm, and 30°C, respectively. The sentence is problematic should be corrected. Does the column correspond to UV280 nm?

What is the DNP?

Novel object recognition test: “On the day of the test, each mouse was replaced in the apparatus in which one of the identical objects was changed with a novel object” What is the kind of novel object mentioned here? Is the novel object completely different from another one in the shape or color etc? These details are missing. It will be much more helpful if the picture is provided showing the setup of Novel object recognition test.

Author Response

Thank you for the valuable comments on this paper. We considered the comments carefully and the manuscript has been revised according to the comments.

In the present work, the authors looked into the effect of PPM, pure TSG and DNP on cognitive behavior (measured by several cognitive related tests) of mice that have been treated with scopolamine. The results indeed clearly showed that the positive effect of improving the cognitive dysfunctions caused by scopolamine. To be accurate, these data only demonstrated that PPM TSG or DNP (at certain doses) can improve the cognitive dysfunctions induced by scopolamine. However, one cannot definitively conclude that PPM could be a functional food and an effective dietary supplement for the prevention and treatment of AD. Such type of conclusion is a bit overstated, thus need to be be revised to a moderate tone. 

; Thank you for the valuable comments. According to your comments, we revised it in the conclusion section (Line 527-530, Page 13).

[Conclusion]

Therefore, we suggest that PPM and TSG improves scopolamine-induced cognitive dysfunction in mice, but further study has to be supported for the clinical application of PPM and TSG against AD prevention and treatment.

Additionally, the authors have not studied the difference in cognitive response between non-treated normal group and only PPM (or TSG, or DNP)-treated group. Such type of experimental results certainly help convince the role of PPM or TSG or DNP in improving the cognitive  behavior, this can be one drawback for well understanding the aim of this work, and thus, the discussion is essentially required.

; We explained in the discussion section (Line 503-513, Page 13).

[Discussion]

In present study, PPM, TSG, and DNP significantly improved cognitive function by regulations of cholinergic dysfunction, BDNF expression, and neuronal apoptosis in the scopolamine-induced cognitive impairment mice. Several previous studies also demonstrated that administration of TSG at dose of 20, 40, 80 mg/kg/day improved cognitive function such as fear memory and novel object recognition by up-regulation of BDNF in normal mice (Chen et al., 2016). In addition, treatment of TSG significantly increased expression of neurotrophic factors including BDNF, nerve growth factor (NGF), and glial cell-derived neurotrophic factor (GDNF) in rat primary astroglial cells (Lin et al., 2016). Furthermore, administration of TSG increased hippocampal synaptic plasticity in normal mice (Wang et al., 2011). These previous studies support the cognitive improvement effects of TSG and PPM under normal mice model as well as AD mice model.

[References]

Chen, T.; Yang, Y.J.; Li, Y.K.; Liu, J.; Wu, P.F.; Wang, F.; Chen, J.G.; Long, L.H. Chronic administration tetrahydroxystilbene glucoside promotes hippocampal memory and synaptic plasticity and activates ERKs, CaMKII and SIRT1/miR-134 in vivo. J. Ethnopharmacol. 2016, 190, 74-82. https://doi.org/10.1016/j.jep.2016.06.012.

Lin, F.; Zhou, Y.; Shi, W.; Wan, Y.; Zhang, Z.; Zhang, F. Tetrahydroxystilbene glucoside improves neurotrophic factors release in cultured astroglia. CNS Neurol. Disord. Drug Targets. 2016, 15, 514-519. https://doi.org/10.2174/1871527314666150821102025.

Wang, T.; Yang, Y.J.; Wu, P.F.; Wang, W.; Hu, Z.L.; Long, L.H.; Xie, N.; Fu, H.; Wang, F.; Chen, J.G. Tetrahydroxystilbene glucoside, a plant-derived cognitive enhancer, promotes hippocampal synaptic plasticity. Eur. J. Pharmacol. 2011, 650, 206-214. https://doi.org/10.1016/j.ejphar.2010.10.002.

Secondly, what is the relative roles of TSG and DNP? Dose the promising therapeutic potential of PPM correlate with TSG or DNP? This is addressed in the discussion.  

; We explained it in discussion section (Line 389-400, Page 10-11; Line 514-520, Page 13).

[Discussion]

DNP was approved under Food and Drug Administration (FDA) drugs for treatment of AD, and it is a well-known treatment agent for AD (Anand and Singh, 2013). DNP acts as a cholinesterase inhibitor by enhancing cholinergic neurotransmission (Anand and Singh, 2013). A number of studies reported that cognitive improvement effects of DNP was related to regulation of cholinergic dysfunction, neuronal apoptosis, and the others, in scopolamine-induced cognitive impairment mice (Hou et al., 2014; Zaki et al., 2014). However, DNP has several side effects such as vomiting, diarrhea, nausea, and muscle cramps in the body (Szeto and Lewis, 2016). Therefore, many studies have been focused on the development of natural products without side effects that can replace synthetic drugs for treatment of AD (Panda and Jhanj, 2020). In this study, we investigated the cognitive improvement effects of PPM and TSG, which is index component of PPM. The DNP was used as a positive control for evaluation of cognitive function.

Taken together with our results, administration of PPM and its index component, TSG, improved cognitive function in the behaviour tests, and they had similar effects to administration of DNP. In addition, PPM- and TSG-administered groups attenuated cholinergic dysfunction, similarly DNP-administered group. Administration of PPM and TSG increased BDNF expression similar to DNP-administered group. Furthermore, PPM- and TSG-administered groups at all concentrations significantly attenuated neuronal apoptosis by reduction of the Bax/Bcl-2 ratio, compared with DNP-administered group.

[References]

Anand, P.; Singh, B.A. Review on cholinesterase inhibitors for Alzheimer’s disease. Arch. Pharm. Res. 2013, 36, 375–399. https://doi.org/10.1007/s12272-013-0036-3

Hou, X.Q.; Wu, D.W.; Zhang, C.X.; Yan, R.; Yang, C.; Rong, C.P.; Zhang, L.; Chang, X.; Su, R.Y.; Zhang, S.J.; He W.Q.; Zhao, Q.; Shi, L.; Su, Z.R.; Chen, Y. B.; Wang, Q.; Fang, S.H. Bushen‑Yizhi formula ameliorates cognition deficits and attenuates oxidative stress‑related neuronal apoptosis in scopolamine‑induced senescence in mice. Int. J. Mol. Med. 2014, 34, 429-439. https://doi.org/10.3892/ijmm.2014.1801

Zaki, H.F.; Abd-El-Fattah, M.A.; Attia, A.S. Naringenin protects against scopolamine-induced dementia in rats. BFPC, 2014, 52, 15-25. https://doi.org/10.1016/j.bfopcu.2013.11.001

Szeto, J.J.Y.; Lewis, J.G.S. (2016). Current treatment options for Alzheimer’s disease and Parkinson’s disease dementia. Curr. Neuropharmacol. 2016, 14, 326-338. https://doi.org/10.2174/1570159X14666151208112754

Panda, S.S.; Jhanji, N. (2020). Natural products as potential anti-Alzheimer agents. Curr. Med. Chem. 2020, 27, 5887-5917. https://doi.org/10.2174/0929867326666190618113613

Minor points:

“To prepare the extract concentrates, each extract supernatant was concentrated to approximately 12 °Bx using a rotary evaporator.” What is the meaning of 12 Bx?

; Thanks for the comment. Brix (Bx) means soluble solid contents

12 °Bx (Brix = soluble solid contents)

For the phytochemical analysis, the flow rate, injection volume, column (XTerra™ RP18, 4.6 × 250 mm, 5 μm, Waters Crop., Milford, MA, USA), and oven temperature were adjusted at 1 mL/min, 20 μL, UV 280 nm, and 30°C, respectively. The sentence is problematic should be corrected. Does the column correspond to UV280 nm?

; Thanks for the good comment. It was rewritten as follows (Line 103-115, Page 3):

The analysis of phytochemical compounds was modified and performed as previously described using a high-performance liquid chromatography (HPLC, Agilent 1260 series, Ag-ilent Co., Santa Clara, CA, USA) methods of Jung et al. (2020). For analyzed the seven phytochemicals, one gram of PPM powders was added the 20 mL of the 50% ethanol and the mixtures was extracted at 70 ± 2˚C for 8 h. The extract samples were filtered through a 0.45-µm membrane filter for analysis of HPLC. Samples were separated on an XTerra RP18 analytical column (4.6 × 250 mm, 5 µm; Waters Corp., Milford, MA, USA). The mobile phase was composed of 0.5% glacial acetic acid in water (solution A) and solvent B (0.5% glacial acetic acid in acetonitrile) using the following gradient program: 0 min 0% B, 10 min 15% B, 15 min 5% B, 20 min 15% B, 25 min 5% B, 30 min 10% B, 40 min 50% B, 45 min 60% B, 60 min 90% B, and 65 min 100% B. The injection volume was 20 µL, and the flow rate was 1.0 mL/min at 30°C (oven temperature). The detection wavelength was UV 280 nm.

[References]

Jung, S.; Son, H.; Hwang, C.E.; Cho, K.M.; Park, S.W.; Kim, H.; Kim, H.J. The root of Polygonum multiflorum Thunb. alleviates non-alcoholic steatosis and insulin resistance in high fat diet-fed mice. Nutrients 2020, 12, 2353. https://doi.org/10.3390/nu12082353.

What is the DNP?

; We explained it in discussion section (Line 389-394, Page 10).

[Discussion]

DNP was approved under Food and Drug Administration (FDA) drugs for treatment of AD, and it is a well-known treatment agent for AD (Anand and Singh, 2013). DNP acts as a cholinesterase inhibitors by enhancing cholinergic neurotransmission (Anand and Singh, 2013). A number of studies reported that cognitive improvement effects of DNP was related to regulation of cholinergic dysfunction, neuronal apoptosis, and the others, in scopolamine-induced cognitive impairment mice (Hou et al., 2014; Zaki et al., 2014).

[References]

Anand, P.; Singh, B.A. Review on cholinesterase inhibitors for Alzheimer’s disease. Arch. Pharm. Res. 2013, 36, 375–399. https://doi.org/10.1007/s12272-013-0036-3

Hou, X.Q.; Wu, D.W.; Zhang, C.X.; Yan, R.; Yang, C.; Rong, C.P.; Zhang, L.; Chang, X.; Su, R.Y.; Zhang, S.J.; He W.Q.; Zhao, Q.; Shi, L.; Su, Z.R.; Chen, Y. B.; Wang, Q.; Fang, S.H. Bushen‑Yizhi formula ameliorates cognition deficits and attenuates oxidative stress‑related neuronal apoptosis in scopolamine‑induced senescence in mice. Int. J. Mol. Med. 2014, 34, 429-439. https://doi.org/10.3892/ijmm.2014.1801

Zaki, H.F.; Abd-El-Fattah, M.A.; Attia, A.S. Naringenin protects against scopolamine-induced dementia in rats. BFPC, 2014, 52, 15-25. https://doi.org/10.1016/j.bfopcu.2013.11.001

Novel object recognition test: “On the day of the test, each mouse was replaced in the apparatus in which one of the identical objects was changed with a novel object” What is the kind of novel object mentioned here? Is the novel object completely different from another one in the shape or color etc? These details are missing. It will be much more helpful if the picture is provided showing the setup of Novel object recognition test.

; We explained information about novel object test in detail in methods section (Line 151-512, Page 4). The method of novel object recognition test was carried out by the method of Barkus et al. (2014).

Figure 1. Design of the novel object recognition test.

[Materials and Methods]

On the day of the test, each mouse was replaced in the apparatus in which one of the identical objects was changed with a novel object, which is a different shape and color from familiar object (Barkus et al., 2014).

[References]

Barkus, C.; Sanderson, D.J.; Rawlins, J.N.P.; Walton, M.E.; Harrison, P.J.; Bannerman, D.M. What causes aberrant salience in schizophrenia? A role for impaired short-term habituation and the GRIA1 (GluA1) AMPA receptor subunit. Mol. Psychiatry 2014, 19, 1060-1070. https://doi.org/10.1038/mp.2014.91

Reviewer 3 Report

In the paper by Kim and colleagues entitled “Protective Effect of Processed Polygoni Multiflori Radix and Its Major Substance during Scopolamine-Induced Cognitive Dysfunction”, the authors evaluate the protective potential of Polygoni multiflori Radix (PPM) and 2,3,5,4′-tetrahydroxystilbene-2-O-β-glucoside (TSG) in a scopolamine-induced amnesia mouse model. While potential protective regimens are of great interest, a poorly characterized model is used. Significantly, more work will be required to properly characterize the study and adequately assess protection. 

  1. In Figure 1, mention the doses for PPM, TSG, DNP, and scopolamine.
  2. For Fig 7 & 8, authors should provide original uncropped and unadjusted blots, including molecular size markers that contributed to the protein quantitation.
  3. Section 2.6: Indicate the anesthesia used to sacrifice the mice.
  4. In a study of cognition, it is important to study brain region(s) related to memory and cognition. The major concern of this reviewer is that the lysate was extracted from the whole “brain tissues” and bears questionable relevance to the studies. It is highly relevant to focus on the hippocampus which is more related to the behavior under study (cognition). Do the authors have a suitable justification for this comment?
  5. The precursor (proBDNF) and its mature form of BDNF preferentially bind specific receptors and exert distinct functions in the brain. Results related to BDNF should be complemented with effects on the pro and mature forms of BDNF. It would be desirable if this study looked into BDNF processing and signaling to evaluate therapeutic applications of Polygoni multiflora.
  6. Additional immunohistochemical analyses are required to be performed to assess cellular phenotype.
  7. The study conclusion is over-interpreted, particularly the claim that “PPM and TSG could be promising candidates as memory enhancers or as therapeutic remedies for cognitive dysfunction in neurodegenerative disorder”.

Author Response

Thank you for the valuable comments on this paper. We considered the comments carefully and the manuscript has been revised according to the comments.

In the paper by Kim and colleagues entitled “Protective Effect of Processed Polygoni Multiflori Radix and Its Major Substance during Scopolamine-Induced Cognitive Dysfunction”, the authors evaluate the protective potential of Polygoni multiflori Radix (PPM) and 2,3,5,4′-tetrahydroxystilbene-2-O-β-glucoside (TSG) in a scopolamine-induced amnesia mouse model. While potential protective regimens are of great interest, a poorly characterized model is used. Significantly, more work will be required to properly characterize the study and adequately assess protection. 

  1. In Figure 1, mention the doses for PPM, TSG, DNP, and scopolamine.

; We added the administration doses of PPM, TSG, and DNP in Figure 1.

  1. For Fig 7 & 8, authors should provide original uncropped and unadjusted blots, including molecular size markers that contributed to the protein quantitation.

; We indicated protein molecular size (kDA) according to protein markers and the blots were replaced to unadjusted bands in Figure 7 and Figure 8.

  1. Section 2.6: Indicate the anesthesia used to sacrifice the mice.

; We explained the anesthesia methods to sacrifice the mice (Line 193, Page 4).

[Materials and Methods]

All mice were anaesthetized with a zoletil50 and rumpun mixture (3:1 ratio) and sacrificed for sample collection after completing the behavioral tests.

  1. In a study of cognition, it is important to study brain region(s) related to memory and cognition. The major concern of this reviewer is that the lysate was extracted from the whole “brain tissues” and bears questionable relevance to the studies. It is highly relevant to focus on the hippocampus which is more related to the behavior under study (cognition). Do the authors have a suitable justification for this comment?

; Thank you for your valuable comment. Several studies also reported the protective effects and mechanisms of agents for AD treatment using whole brain, although hippocampus is highly related to cognitive function (Dong et al., 2009; Li et al., 2012). Scopolamine-induced mice showed cholinergic dysfunction by up-regulation of AChE and down-regulation of ACh in the whole brain (Tung et al., 2017). In addition, scopolamine-treated mice decreased BDNF expression and induced neuronal apoptosis by increase of the Bax/Bcl-2 ratio in whole brain tissue (Park et al., 2019). Previous studies examined cognitive improvement effects and its mechanisms of natural products and active compounds such as Emilia coccinae and naringenin in the whole brain tissue of mouse (Foyet et al., 2015; Zaki et al., 2014). On the other hand, several studies reported that administration of TSG enhanced cognitive and memory function by increase of BDNF expression in the hippocampus of mice (Chen et al., 2016; Chen et al., 2017; Wang et al., 2017). Therefore, we suggest that TSG-administered mice would play a protective role from cognitive impairment by up-regulation of BDNF in hippocampus.

[References]

Dong, H.; Yuede, C.M.; Coughlan, C.A.; Murphy, K.M.; Csernansky, J.G. Effects of donepezil on amyloid-β and synapse density in the Tg2576 mouse model of Alzheimer's disease. Brain Res. 2009, 1303, 169-178. https://doi.org/10.1016/j.brainres.2009.09.097

Li, Q.; Chen, M.; Liu, H.; Yang, L.; Yang, G. Expression of APP, BACE1, AChE and ChAT in an AD model in rats and the effect of donepezil hydrochloride treatment. Mol. Med. Rep. 2012, 6, 1450-1454. https://doi.org/10.3892/mmr.2012.1102.

Tung, B.T.; Hai, N.T.; Thu, D.K. Antioxidant and acetylcholinesterase inhibitory activities in vitro of different fraction of Huperzia squarrosa (Forst.) Trevis extract and attenuation of scopolamine-induced cognitive impairment in mice. J. Ethnopharmacol. 2017, 198, 24-32. https://doi.org/10.1016/j.jep.2016.12.037

Park, J.W.; Kim, J.E.; Kang, M.J.; Choi, H.J.; Bae, S.J.; Kim, S.H.; Jung, Y.S. Hong, J.T. Hwang, D.Y. Anti-oxidant activity of gallotannin-enriched extract of Galla rhois can associate with the protection of the cognitive impairment through the regulation of BDNF signaling pathway and neuronal cell function in the scopolamine-treated ICR mice. Antioxidants, 2019, 8, 450. https://doi.org/10.3390/antiox8100450

Foyet, H.S.; Abaïssou, H.H.N.; Wado, E.; Acha, E.A.; Alin, C. Emilia coccinae (SIMS) G Extract improves memory impairment, cholinergic dysfunction, and oxidative stress damage in scopolamine-treated rats. BMC Compl. Alternative Med. 2015, 15, 1-12.

Zaki, H.F.; Abd-El-Fattah, M.A.; Attia, A.S. Naringenin protects against scopolamine-induced dementia in rats. BFPC, 2014, 52, 15-25. https://doi.org/10.1016/j.bfopcu.2013.11.001

Chen, T.; Yang, Y.J.; Li, Y.K.; Liu, J.; Wu, P.F.; Wang, F.; Chen, J.G.; Long, L.H. Chronic administration tetrahydroxystilbene glucoside promotes hippocampal memory and synaptic plasticity and activates ERKs, CaMKII and SIRT1/miR‐134 in vivo. J. Ethnopharmacol. 2016, 190, 74‐82. https://doi.org/10.1016/j.jep.2016.06.012,

Chen, Z.; Huang, C.; He, H.; Ding, W. 2,3,5,4'-Tetrahydroxystilbene-2-O-β-D-glucoside prevention of lipopolysaccharide-induced depressive-like behaviors in mice involves neuroinflammation and oxido-nitrosative stress inhibition. Behav. Pharmacol. 2017, 28, 365-374. https://doi.org/10.1097/FBP.0000000000000307.

Wang, H.; Zhao, Y.; Wang, Y.J.; Song, L.; Wang, J.L.; Huang, C.; Zhang, W.; Jiang, B. Antidepressant-like effects of tetrahydroxystilbene glucoside in mice: Involvement of BDNF signaling cascade in the hippocampus. CNS Neurosci. Ther. 2017, 23, 627-636. https://doi.org/10.1111/cns.12708.

  1. The precursor (proBDNF) and its mature form of BDNF preferentially bind specific receptors and exert distinct functions in the brain. Results related to BDNF should be complemented with effects on the pro and mature forms of BDNF. It would be desirable if this study looked into BDNF processing and signaling to evaluate therapeutic applications of Polygoni multiflora.

; We explained it in discussion section (Line 474-492, Page 12).

[Discussion]

The BDNF exists two forms such as the precursor (pro BDNF) and its mature form of BDNF in the brain. The precursor BDNF preferentially binds p75 neurotrophin receptor (NTR), and then it is converted to mature BDNF by extracellular proteases (Zhang et al., 2016). The mature BDNF preferentially binds to tropomyosine-related kinase B (TrkB) receptor (Zhang et al., 2016). Pro BDNF binds to p75 NTR and then activates apoptotic pathways in neurons and glial, and it negatively regulates neuronal remodeling and synaptic plasticity (Meng et al., 2014; Budni et al., 2015). On the other hand, mature BDNF enhances long-term potentiation and it is critical for neuroplasticity and neuronal function in the brain (Meng et al., 2014). In the AD patients, the both pro BDNF and mature BDNF were decreased in brain (Peng et al., 2005). In our study, administered-PPM or TSG group elevated protein levels of mature BDNF in the brain compared with scopolamine-induced control group, indicating that PPM and TSG improve long-term memory and neuronal function under cognitive impairment induced by scopolamine. In addition, according to previous study, TSG-administered mice increased BDNF as well as TrkB expression in the brain (Wang et al., 2017). Previous study investigated effects of PM on pro BDNF and mature BDNF expression in neurotoxicity-induced hippocampal neuronal cells (Ahn et al., 2015). Treatment of PM significantly increased mature BDNF expression (Ahn et al., 2015). Therefore, PPM and TSG could activate the mature form of BDNF, but the further study on BDNF processing and signaling of PPM or TSG has to be supported for the application of AD therapy.

[References]

Zhang, J.C.; Yao, W.; Hashimoto, K. Brain-derived neurotrophic factor (BDNF)-TrkB signaling in inflammation-related depression and potential therapeutic targets. Curr. Neuropharmacol. 2016, 14, 721-731. https://doi.org/10.2174/1570159x14666160119094646.

Meng, Y.; Chopp, M.; Zhang, Y.; Liu, Z.; An, A.; Mahmood, A.; Xiong, Y. Subacute intranasal administration of tissue plasminogen activator promotes neuroplasticity and improves functional recovery following traumatic brain injury in rats. PLoS One. 2014, 9, e106238. https://doi.org/10.1371/journal.pone.0106238.

Budni, J.; Bellettini-Santos, T.; Mina, F.; Garcez, M.L.; Zugno, A.I. The involvement of BDNF, NGF and GDNF in aging and Alzheimer's disease. Aging Dis. 2015, 6, 331-341. https://doi.org/10.14336/AD.2015.0825.

Peng, S.; Wuu, J.; Mufson, E.J.; Fahnestock, M. Precursor form of brain-derived neurotrophic factor and mature brain-derived neurotrophic factor are decreased in the pre-clinical stages of Alzheimer's disease. J. Neurochem. 2005, 93, 1412-1421. https://doi.org/10.1111/j.1471-4159.2005.03135.x.

Wang, H.; Zhao, Y.; Wang, Y.J.; Song, L.; Wang, J.L.; Huang, C.; Zhang, W.; Jiang, B. Antidepressant-like effects of tetrahydroxystilbene glucoside in mice: Involvement of BDNF signaling cascade in the hippocampus. CNS Neurosci. Ther. 2017, 23, 627-636. https://doi.org/10.1111/cns.12708.

Ahn, S.M.; Kim, Y.R.; Kim, H.N.; Shin, H.K.; Choi, B.T. (2015). Beneficial effects of Polygonum multiflorum on hippocampal neuronal cells and mouse focal cerebral ischemia. Am. J. Chin. Med. 2015, 43, 637-651. https://doi.org/10.1142/S0192415X15500391

  1. Additional immunohistochemical analyses are required to be performed to assess cellular phenotype.

; According to previous several studies, effects of TSG on histochemical changes were reported in the brain using immunohistochemical analysis. Administration of TSG up-regulated the expression of Klotho protein, a common anti-aging protein, in the brain of aging mice (Zhou et al., 2013). In the Glogi staining, administration of TSG increased dendritic spine density, which is hall marker of memory storage and synaptic transmission and number of possible contacts between neurons, in the brain (Chen et al., 2016). TSG attenuated neuronal cell loss by inhibition of inflammation-related factors such as TH-positive DA neurons, OX-42-positive microglia, GFAP-positive astroglia in the lipopolysachharide-induced brain (Zhou et al., 2018). However, further study on effects of PPM and TSG on histochemical changes in the scopolamine-induced cognitive impairment mouse using immunohistochemical analyses has to be needed.

[Referneces]

Chen, T.; Yang, Y.J.; Li, Y.K.; Liu, J.; Wu, P.F.; Wang, F.; Chen, J.G.; Long, L.H. Chronic administration tetrahydroxystilbene glucoside promotes hippocampal memory and synaptic plasticity and activates ERKs, CaMKII and SIRT1/miR‐134 in vivo. J. Ethnopharmacol. 2016, 190, 74‐82. https://doi.org/10.1016/j.jep.2016.06.012,

Zhou, Y.; Wang, G.; Li, D.; Wang, Y.; Wu, Q.; Shi, J.; Zhang, F. Dual modulation on glial cells by tetrahydroxystilbene glucoside protects against dopamine neuronal loss. J. Neuroinflammation. 2018, 15, 161. https://doi.org/10.1186/s12974-018-1194-5.

Zhou, X.X.; Yang, Q.; Xie, Y.H.; Sun, J.Y.; Qiu, P.C.; Cao, W.; Wang, S.W. Protective effect of tetrahydroxystilbene glucoside against D-galactose induced aging process in mice. Phytochem. Lett. 2013, 6, 372-378. https://doi.org/10.1016/j.phytol.2013.05.002

  1. The study conclusion is over-interpreted, particularly the claim that “PPM and TSG could be promising candidates as memory enhancers or as therapeutic remedies for cognitive dysfunction in neurodegenerative disorder”.

; Thank you for the valuable comments. According to your comments, we revised it in the conclusion section (Line 527-530, Page 13).

[Conclusion]

Therefore, we suggest that PPM and TSG improves scopolamine-induced cognitive dysfunction in mice, but further study has to be supported for the clinical application of PPM and TSG against AD prevention and treatment.

Round 2

Reviewer 3 Report

The reviewer appreciates the authors' effort in responding to the comments. However, still there are several concerns 

  1. The authors have inadequately addressed the major concern #2 raised by this reviewer. The revised version does not provide the original uncropped and unadjusted blots that contributed to the protein quantitation in this paper. It is highly possible that the quantitation has been performed using several blots/gels. 
  2. The authors' response that several other studies reporting the protective effects and mechanisms of agents for AD treatment using whole-brain are not convincing. It is required to be justified why molecular assessments were not specifically performed with hippocampus tissue in the present study.
  3. There are grammatical mistakes in the text. The paper needs a thorough revision by a native English speaker.

Author Response

Thank you for the valuable comments on this paper. We attached the response as a Word file. 

Round 3

Reviewer 3 Report

The authors' response that the amount of hippocampus is not sufficient to evaluate all the reported outcome measures in the paper is incomprehensive and unjustifiable. 

Author Response

Thank you for the valuable comments on this paper. We considered the comments carefully and the manuscript has been revised according to the comments.

The authors' response that the amount of hippocampus is not sufficient to evaluate all the reported outcome measures in the paper is incomprehensive and unjustifiable.

; Several studies reported that cholinergic dysfunction, BDNF expression, and neuronal apoptosis are related to the various area of brain including hippocampus. Cholinergic dysfunction affects to various areas of brain including the cortex, the entorhinal area, the hippocampus, the ventral striatum and the basal part of the forebrain, and the others (Kása et al., 1997; Li et al., 2018). In addition, cognitive impairment mice decreased ACh level and increased AChE activity in cerebral cortex, hypothalamus, and whole brain tissue (Wang et al., 2018; You et al., 2020). BDNF is expressed in the various parts of brain such as hippocampus, amygdala, projection areas of the olfactory system, inner and outer pyramidal layers of the neocortex, claustrum, cerebellum and the superior colliculus (Connor et al., 1997). In AD, the expression of BDNF is down-regulated in the hippocampus, temporal cortex, and prefrontal cortex of brain, thereby BDNF is significantly associated with hippocampal and/or whole brain (Connor et al., 1997; Aarons et al., 2019; Honea et al., 2013). Furthermore, in various areas of brain such as hippocampus and temporal cortex, neuronal apoptosis was observed by inhibition of Bax/Bcl-2 ratio in the AD (Kitamura et al., 1998). In particular, cognitive impairment mouse model up-regulated Bax and down-regulated Bcl-2 expressions in the whole brain tissue (Yu et al., 2019). Therefore, we investigated the protective mechanisms of PPM and TSG against cognitive impairment induced by scopolamine in the whole brain tissue.

[References]

Aarons, T.; Bradburn, S.; Robinson, A.; Payton, A.; Pendleton, N.; Murgatroyd, C. Dysregulation of BDNF in prefrontal cortex in Alzheimer’s disease. Journal of Alzheimer's Disease, 2019, 69, 1089-1097.

Connor, B.; Young, D.; Yan, Q.; Faull, R.L.M.; Synek, B.; Dragunow, M. Brain-derived neurotrophic factor is reduced in Alzheimer's disease. Molecular Brain Research, 1997, 49, 71-81.

Honea, R.A.; Cruchaga, C.; Perea, R.D.; Saykin, A.J.; Burns, J.M.; Weinberger, D.R.; Goate, A.M. Alzheimer’s Disease Neuroimaging Initiative (ADNI). Characterizing the role of brain derived neurotrophic factor genetic variation in Alzheimer's disease neurodegeneration. PLoS One. 2013, 8, e76001.

Kása, P.; Rakonczay, Z.; Gulya, K. The cholinergic system in Alzheimer's disease. Progress in Neurobiology, 1997, 52, 511-535.

Kitamura, Y.; Shimohama, S.; Kamoshima, W.; Ota, T.; Matsuoka, Y.; Nomura, Y.; Smith, M.A.; Perry, G.; Whitehouse, P.J.; Taniguchi, T. Alteration of proteins regulating apoptosis, Bcl-2, Bcl-x, Bax, Bak, Bad, ICH-1 and CPP32, in Alzheimer's disease. Brain Research, 1998, 780, 260-269.

Li, X.; Yu, B.; Sun, Q.; Zhang, Y.; Ren, M.; Zhang, X.; Li, A.; Yuan, J.; Madisen, L.; Luo, Q.; Zeng, H.; Gong, H.; Qiu, Z. Generation of a whole-brain atlas for the cholinergic system and mesoscopic projectome analysis of basal forebrain cholinergic neurons. Proceedings of the National Academy of Sciences, 2018, 115, 415-420.

Wang, C.; Cai, X.; Hu, W.; Li, Z.; Kong, F.; Chen, X.; Wang, D. Investigation of the neuroprotective effects of crocin via antioxidant activities in HT22 cells and in mice with Alzheimer's disease. International Journal of Molecular Medicine, 2019, 43, 956-966.

You, S.H.; Jang, M.; Kim, G.H. Mori cortex Radicis attenuates high fat diet-induced cognitive impairment via an IRS/Akt signaling pathway. Nutrients 2020, 12, 1851.

Yu, H.; Yuan, B.; Chu, Q.; Wang, C.; Bi, H. Protective roles of isoastilbin against Alzheimer's disease via Nrf2‑mediated antioxidation and anti‑apoptosis. International journal of molecular medicine 2019, 43, 1406-1416.
